# Treatment of Colorectal Cancer in Certified Centers: Results of a Large German Registry Study Focusing on Long-Term Survival

**DOI:** 10.3390/cancers15184568

**Published:** 2023-09-15

**Authors:** Vinzenz Völkel, Michael Gerken, Kees Kleihues-van Tol, Olaf Schoffer, Veronika Bierbaum, Christoph Bobeth, Martin Roessler, Christoph Reissfelder, Alois Fürst, Stefan Benz, Bettina M. Rau, Pompiliu Piso, Marius Distler, Christian Günster, Judith Hansinger, Jochen Schmitt, Monika Klinkhammer-Schalke

**Affiliations:** 1Tumor Center Regensburg, Center of Quality Management and Health Services Research, University of Regensburg, 93053 Regensburg, Germanymonika.klinkhammer-schalke@klinik.uni-regensburg.de (M.K.-S.); 2Bavarian Cancer Registry, Regional Center Regensburg, Bavarian Health and Food Safety Authority, 93053 Regensburg, Germany; 3Arbeitsgemeinschaft Deutscher Tumorzentren e.V. (ADT), 14057 Berlin, Germany; 4Center for Evidence-Based Healthcare (ZEGV), Faculty of Medicine, University Hospital Carl Gustav Carus and Carl Gustav Carus, Dresden University of Technology (TU Dresden), 01307 Dresden, Germany; olaf.schoffer@ukdd.de (O.S.); jochen.schmitt@ukdd.de (J.S.); 5Department of Surgery, Universitätsmedizin Mannheim, Medical Faculty Mannheim, Heidelberg University, 68167 Mannheim, Germany; 6Klinik für Allgemein-, Viszeral-, Thoraxchirurgie und Adipositasmedizin, Caritas Krankenhaus St., 93053 Regensburg, Germany; 7Klinik für Allgemein-, Viszeral-, Thorax- und Kinderchirurgie, 71032 Böblingen, Germany; 8Department of General, Visceral and Thoracic Surgery, Hospital of Neumarkt, 92318 Neumarkt in der Oberpfalz, Germany; 9Klinik für Allgemein- und Viszeralchirurgie, Krankenhaus der Barmherzigen Brüder, 93049 Regensburg, Germany; 10Department of Visceral, Thoracic and Vascular Surgery, University Hospital Carl Gustav Carus Dresden, Faculty of Medicine Carl Gustav Carus, Dresden University of Technology (TU Dresden), 01307 Dresden, Germany; 11WIdO—AOK Research Institute, 10178 Berlin, Germany

**Keywords:** certified cancer center, colon cancer, rectal cancer, cohort study, registries, survival, quality of cancer care, evidence-based medicine, WiZen, German Cancer Society

## Abstract

**Simple Summary:**

Certification in oncology aims to establish structural and procedural standards according to evidence-based guidelines. The WiZen study is the largest study so far to analyze the effect of the certification of designated cancer centers on survival in Germany. Based on clinical cancer registry data of 47.440 colorectal cancer patients treated between 2009 and 2017, the present study shows that treatment at colorectal cancer centers has been associated with significantly better outcomes. Patients treated at certified facilities had an eleven percent (colon)/nine percent (rectum) lower risk of dying within the first five years after diagnosis. These findings support the shift towards a more structured cancer care system.

**Abstract:**

(1) Background: The WiZen study is the largest study so far to analyze the effect of the certification of designated cancer centers on survival in Germany. This certification program is provided by the German Cancer Society (GCS) and represents one of the largest oncologic certification programs worldwide. Currently, about 50% of colorectal cancer patients in Germany are treated in certified centers. (2) Methods: All analyses are based on population-based clinical cancer registry data of 47.440 colorectal cancer (ICD-10-GM C18/C20) patients treated between 2009 and 2017. The primary outcome was 5-year overall survival (OAS) after treatment at certified cancer centers compared to treatment at other hospitals; the secondary endpoint was recurrence-free survival. Statistical methods included Kaplan–Meier analysis and multivariable Cox regression. (3) Results: Treatment at certified hospitals was associated with significant advantages concerning 5-year overall survival (HR 0.92, 95% CI 0.89, 0.96, adjusted for a broad range of confounders) for colon cancer patients. Concentrating on UICC stage I–III patients, for whom curative treatment is possible, the survival benefit was even larger (colon cancer: HR 0.89, 95% CI 0.84, 0.94; rectum cancer: HR 0.91, 95% CI 0.84, 0.97). (4) Conclusions: These results encourage future efforts for further implementation of the certification program. Patients with colorectal cancer should preferably be directed to certified centers.

## 1. Introduction

Colorectal cancer is one of the most common malignancies worldwide [1,2]. In Germany, 5.3% of women and 6.7% of men will be diagnosed with colorectal cancer in the course of their life; this corresponds to a national incidence of approximately 60,000 diagnoses per year [3]. With these figures, colorectal cancer belongs to the three most frequent tumor diseases in Germany. The observed survival rate after five years for all stages has been constantly improving over the past decades and amounts to 54% among female and 52% among male patients [3]. Depending on the tumor location, Union for International Cancer Control (UICC) stage, and therapy intention, the treatment of colorectal cancer relies on surgical resection in locally advanced and metastatic cases combined with pre- or postoperative chemo-, radio-, or targeted therapy [4].

Health care systems worldwide aim to improve cancer care quality by means of the accreditation or certification of specialized hospitals [5,6,7,8]. To promote optimal, guideline-based therapy pathways, a certification program for treating facilities has been established in Germany. Since 2003, the German Cancer Society (GCS; German: Deutsche Krebsgesellschaft, DKG) has offered organ-specific certification programs [9,10,11]. As of today, there exist 18 different GCS certification programs and 1402 GCS-certified centers [12]. Of those, 314 are specialized in colorectal cancer [13]. To obtain GCS certification, a hospital has to fulfill a broad variety of requirements. They include structural measures concerning, e.g., regular interdisciplinary communication and consensus decision making in structured tumor boards, or the maintenance of multi-professional outreach networks. Moreover, they have to report about 30 performance indicators reflecting process and outcome quality in compliance with official treatment guidelines (e.g., minimal annual caseloads of 30 colon and 20 rectum cancer resections per certified center and year, share of therapy pathways deviating from tumor board decisions) [14]. Certified colorectal cancer centers must undergo regular external audits and their performance indicators are included in publicly available annual quality reports published by the GCS [15].

In 2018, a registry-based health service analysis showed that treatment at GCS-certified colorectal cancer facilities might be associated with significant survival benefits [16]. However, this study and a few other analyses for different tumor entities from other countries [17,18,19,20,21] were subject to some limitations (e.g., concerning the coverage area, transferability from other health care systems to Germany), leaving room for discussion. The WiZen study (German Innovation Fund, grant number 01VSF17020) currently represents the largest study on the topic. The present publication aims to provide an in-depth overview of the study’s specific results regarding colorectal cancer with a special focus on clinical cancer registry-based analyses (the statutory health insurance-based analyses will be published elsewhere).

## 2. Materials and Methods

### 2.1. Aim

The WiZen study has been jointly conducted by four different institutions with expertise in clinical epidemiology and evidence-based medicine: Zentrum für Evidenzbasierte Gesundheitsversorgung (ZEGV)/Hochschulmedizin Dresden, Germany, Tumorzentrum Regensburg (TZR), Germany, Arbeitsgemeinschaft Deutscher Tumorzentren e. V. (ADT), Berlin, Germany and Wissenschaftliches Institut der AOK (WIdO), Berlin, Germany. Cooperation partners who provided relevant data from cancer registries and the certification program were GCS, Berlin, Germany. Klinisches Krebsregister Dresden (KKRD), Germany, Klinisches Krebsregister Erfurt (KKRE), Germany, and Klinisches Krebsregister für Brandenburg und Berlin (KKRBB), Germany.

The WiZen study has been designed as a set of retrospective cohort studies aiming to evaluate whether treatment in a GCS-certified cancer center is associated with better overall survival. Besides colorectal cancer, it focuses on seven other tumor entities (breast cancer, gynecological cancer, head and neck cancer, lung cancer, neurooncological tumors, pancreatic cancer, and prostate cancer; a general report of the project results is available online [22]).

### 2.2. Data Source

The findings reported in this publication are based on a comprehensive dataset provided by four large clinical cancer registries (TZR, KKRD, KKRE, KKRBB). These cancer registries fulfill an official mandate and collect data on all cancer patients registered in their catchment area. The dataset contained demographic characteristics (age, sex, date of death), detailed tumor characteristics (date of diagnosis, histological subtype, tumor stage according to the International Union against Cancer, UICC, tumor grade, lymphatic and venous invasion), as well as information about treatment procedures. It covered an observation period from 2006 to 2017.

### 2.3. Inclusion and Exclusion Criteria

The following criteria were used to define the population for the analyses shown in this paper:(a)Diagnosis of colorectal cancer according to the ICD-10-GM codes C18 (malignant neoplasm of the colon) or C20 (malignant neoplasm of the rectum).(b)Age of at least 18 years at the time of diagnosis.(c)No previous diagnoses of colorectal cancer (a patient was only considered as incident between 2009 and 2017 if there were no earlier diagnoses of colorectal cancer recorded; to avoid issues with missing information concerning earlier tumor diagnoses, patients with a cancer diagnosis between 2006 and 2008 were excluded a priori following the guideline “good practice of secondary data analysis” [23] since it would not have been possible to assess a case’s compliance to this inclusion criterion). Previous diagnoses of other, non-colorectal cancer were no exclusion criterion.(d)Sufficient information concerning the certification status of the treating hospital (in this context, treatment at a non-certified institution that belongs to an association containing a GCS-certified colorectal cancer center was also considered a certified center treatment).(e)Consistent histological subtype (only adenocarcinoma, exclusion of, e.g., lymphoma or sarcoma).

### 2.4. Statistical Analysis

The primary outcome was overall survival up to five years after diagnosis, and the secondary outcome was 5-year recurrence-free survival. Each included patient was considered to be at risk of death or tumor recurrence from the date of diagnosis onwards. The follow-up period ended at the date of death or tumor recurrence, or on 31 December 2017, in the absence of an event. To compare the unadjusted survival rates between GCS-certified colorectal cancer centers and non-certified hospitals, the Kaplan–Meier method was employed. To adjust for a variety of important confounders (age, sex, year of diagnosis, UICC stage, grade, lymphatic and venous invasion), multivariable Cox regression models were developed. All significance tests were two-sided with a significance level of 0.05. All reported results are presented together with the corresponding *p*-value and/or the upper and the lower border of the 95% confidence interval. The analyses were performed with IBM SPSS 25 (IBM SPSS Statistics for Windows, Version 25.0. Armonk, NY, USA: IBM Corp.).

### 2.5. Data Protection and Ethics

The data were pseudonymized at the participating cancer registries. The pseudonymized data were analyzed at TZR. The WiZen study was approved by the ethics committee of the TU Dresden (approval number: EK95022019). The study was registered at ClinicalTrials.gov (identifier: NCT04334239). The data processing and analyses were conducted in line with the Declaration of Helsinki and the General Data Protection Regulation of the European Union.

## 3. Results

### 3.1. Inclusion Process

Between 2009 and 2017, the dataset contained 30,742 patients with colon (ICD-10-GM C18) and 17,040 with rectum cancer (ICD-10-GM C20). After the application of all inclusion criteria, 30,497 (99.2%), and 16,943 (99.4%) patients, respectively, were used for the projected analyses.

### 3.2. Share of Patients Treated in GCS-Certified Colorectal Cancer Centers

The share of patients treated in GCS-certified colorectal cancer centers was 27.9% (colon; rectum: 31.7%) at the beginning of the observation period in 2009 and increased to 51.9% (colon; rectum: 49.4%) in 2016; thereafter, it slightly dropped (Figure 1).

### 3.3. Description of Collectives

The colon cancer patients treated in GCS-certified centers showed a similar sex distribution to the patients from non-certified hospitals. Among the rectum cancer patients, the share of male patients was only slightly higher in non-certified hospitals (66.5% vs. 62.9%). Concerning the age distribution, no relevant differences between centers and other hospitals were seen, either. The colon and rectum cancer patients treated at certified centers suffered from advanced UICC stages (III and IV) more often; moreover, an unknown tumor stage was seen less frequently in center patients (colon: 6.7% vs. 17.5%; rectum: 6.9% vs. 16.2%). More details can be found in Table 1 and Table 2.

### 3.4. Survival Analyses

The mean follow-up—estimated by means of the reverse Kaplan–Meier method—was 3.39 years in the complete cohort (95% CI 3.36–3.42) and the median follow-up was 3.20 (95% CI 3.14–3.25). In patients treated at certified centers, the mean follow-up was 3.57 years (95% CI 3.53–3.61; median 3.51, 95% CI 3.44–3.58) and in patients treated at non-certified hospitals was 3.23 years (95% CI 3.18–3.28; median 2.84, 95% CI 2.75–2.93), respectively.

#### 3.4.1. Overall Survival

For colon cancer patients of GCS-certified centers, the 5-year Kaplan–Meier survival rate over all stages was 45% compared to 39% for patients from other non-certified hospitals (Figure 2a). The difference between the two survival curves was highly significant (*p* < 0.001). Moreover, 48% of the rectum cancer patients treated in GCS-certified centers were still alive after five years; among patients of other hospitals, this estimated rate was 41%. Again, the difference between the two Kaplan–Meier survival curves was highly significant (*p* < 0.001, Figure 2b).

For colon cancer patients, the unadjusted hazard ratio over all patients for all-cause mortality was 0.868 (95% CI 0.837, 0.900) for treatment in a GCS-certified cancer center compared to other hospitals. Adjusting for age, sex, year of diagnosis, UICC stage, grade, and lymphatic and venous invasion, it changed to 0.921 (95% CI 0.887, 0.956, Figure 3). The results for all covariates contained in the adjusted Cox regression model can be found in Appendix A. For rectal cancer patients, the corresponding hazard ratios were 0.820 (unadjusted, 95% CI 0.780, 0.862) and 0.978 (adjusted, 95% CI 0.929, 1.029), respectively. For colon cancer, this indicates a moderate, yet significant superiority concerning overall survival for treatment in a GCS-certified colorectal cancer center (*p* < 0.001), while no significant difference was seen for rectum cancer patients of all stages combined.

Excluding patients with unknown UICC stage, the adjusted hazard ratio for all-cause mortality changed to 0.908 (colon cancer, 95% CI 0.873, 0.945) and 0.961 (rectum cancer, 95% CI 0.909, 1.014) in favor of treatment in GCS-certified cancer centers (Appendix A). Analyzing UICC stage I to III patients only, the effect of treatment in a GCS-certified center increased (colon cancer: HR 0.889; 95% CI 0.840, 0.940) and became significant for rectum cancer, too (HR 0.905, 95% CI 0.841, 0.973, Figure 3).

For both tumor locations, the effect of center treatment was stronger at lower (colon cancer: UICC I: HR 0.876, UICC II: HR 0.867; rectum cancer: UICC I: HR 0.716, UICC II: HR 0.933, Figure 3) than in higher stages (colon cancer: UICC III: HR 0.907, UICC IV: HR 0.936; rectum cancer: UICC III: HR 0.956, UICC IV: HR 1.049, Figure 3).

#### 3.4.2. Recurrence-Free Survival

For R0-resected UICC stage I–III patients, it was also possible to analyze recurrence-free survival. For colon cancer patients treated at GCS-certified centers, the 5-year recurrence-free survival rate was 61% compared to 55% for patients from non-certified hospitals (*p* < 0.001, Figure 4a). For rectal cancer patients, the 5-year recurrence-free survival rates were 62% and 54%, respectively (*p* < 0.001, Figure 4b).

The adjusted hazard ratio for death or tumor recurrence was 0.878 (colon cancer, 95% CI 0.832, 0.927) and 0.856 (rectum cancer, 95% CI 0.796, 0.921, Table 3), indicating the significant superiority of center treatment.

## 4. Discussion

### 4.1. Summary

Since 2003, the German Cancer Society has certified hospitals specializing in the treatment of cancer and fulfilling certain quality standards. Currently, there exist 314 designated colorectal cancer centers in Germany [13], but evidence as to whether treatment at certified facilities is associated with better survival is scarce. With 47,440 included patients, the analyses presented in this publication represent the largest registry-based study on the topic.

After adjustment for important covariates like age, tumor stage, and the year of index treatment, the hazard of death was significantly lower by 7.9 percent in colon cancer patients treated in certified centers (HR 0.921), whereas only a small and non-significant hazard reduction was seen in rectum cancer patients (HR 0.978). If only patients in non-metastatic stages (UICC I–III) were analyzed, the hazard ratio for center treatment decreased to 0.889 in colon and 0.905 in rectum cancer patients, indicating a substantial and significant survival benefit for both tumor entities.

### 4.2. Effect of Certification

In oncology, great efforts are taken to achieve rather modest improvements of prognosis. In comparison to this, a broader implementation of certification programs and concentration of cancer treatment in certified institutions is associated with larger survival benefits and lower costs for the health care system. Slightly longer traveling distances to the next certified colorectal cancer center (in over 50% of the cases, 20 min or less [24]) seem a rather modest price for an overall survival benefit of up to 28 percent, depending on the UICC stage. Moreover, previous studies show that treatment at GCS-certified colorectal cancer centers is associated with significantly lower treatment costs, even if additional expenses induced by certification requirements are taken into account [25].

The findings presented in this paper are in concordance with other analyses of the WiZen study based on statutory health insurance data [22]. For colon cancer, the adjusted hazard ratio for treatment in certified centers was identical, underlining the high external validity of the presented results. Moreover, the results of the present study support the findings of earlier studies on the topic, which were limited by smaller sample sizes and locally restricted study cohorts. Using clinical cancer registry data from a southern German region with 1.1 million inhabitants from 2004 to 2013, Völkel et al. [16] observed a significant overall survival benefit (HR 0.81) for center treatment. Trautmann et al. analyzed statutory health insurance data from 2005 to 2015 from Saxony, Germany, and also reported a significant survival benefit for patients treated in certified hospitals (OAS: HR 0.90, [26]). Similar results can be found in the international literature, although a comparison to differently structured health care systems and specific interventions within these systems is always difficult. In 2008, Paulson et al. [27] retrospectively analyzed more than 40,000 patients from the US having received surgery for colon and rectum carcinoma. The postoperative mortality in National Cancer Institute (NCI)-designated centers was lower (3.2% vs. 6.7%); additionally, they observed a significant long-term survival benefit for colon (HR 0.84) and rectum cancer patients (HR 0.85), which is comparable to the results presented in this study. In 2021, Okawa et al. [28] published a registry-based observational study about the health care system in Japan, where accredited, high-capacity, highly experienced cancer care hospitals also exist; the findings showed that treatment in these intuitions is associated with higher adjusted all-site 3-year survival rates (86.6%) compared to non-designated hospitals (78.8%).

The fact that the survival benefits observed in the present study were particularly high in UICC stage I–III patients is hardly surprising; for these patients, clear guideline recommendations and quality standards for surgery and (neo-)adjuvant procedures [4] are provided and implemented. Correspondingly, GCS certification strongly focuses on guideline-adherent therapy pathways and is more than a simple volume-based centralization process. Concerning the predominantly palliative UICC stage IV patients, it has to be acknowledged that prolongation of survival is not always the primary treatment aim [29,30]. Future studies on this topic for this specific patient group should implement additional outcome variables reflecting aspects like quality of life and the patients’ perspective.

### 4.3. Strengths and Limitations

For ethical and practical reasons, it is impossible to conduct a randomized controlled trial to analyze the effectiveness of center-based cancer treatment [31,32,33]. Patients are free to present to their hospital of choice. Usually, their decision incorporates factors like regional accessibility, patient mobility, referral by other health professionals, advice from other patients, and many more [34]. Subjecting cancer patients to a random allocation process would impair their right of self-determination. Consequently, an observational study design with independent standardized controlled prospective data collection represents the most adequate methodology. Using population-based clinical cancer registry data of four large German cancer registries covering different regions, the results of this study are highly reliable and representative. Every patient with colorectal cancer registered within the catchment area of these cancer registries is part of the truly population-based study collective. No patient was excluded due to unfavorable characteristics like high age or advanced tumor stages [35,36].

With observational data and non-randomized group allocation, adjustment for potential confounders is crucial. Making use of a comprehensive dataset, it was possible to include a broad range of demographic and disease-specific items in the multivariable regression analyses. For certain variables like, e.g., tumor stage, information was partially missing—this was more often the case in patients from non-certified hospitals, which might be a consequence of lower documentation standards in these hospitals. However, sensitivity analyses with the inclusion and exclusion of patients with incomplete information were performed and the results remained stable.

The strength of the cancer registry-based part of the WiZen analyses presented in our manuscript consists of the detailed information about tumor characteristics. Nevertheless, it would have been desirable to include additional confounders like teaching status or hospital volume. However, for some of the cases, no documentation about the specific institution a case was treated in was available. For these cases, certification status was retrieved from a generic variable “center treatment yes/no”. Notwithstanding this, it was possible to adjust for these factors in separate analyses of the WiZen project, which were based on statutory health insurance data and that will be published elsewhere. In these analyses, one can see that hospital size and teaching status are seen more often among certified centers; these factors did indeed have a significant influence on survival, but even after adjustment for these factors, certification as colorectal cancer center was associated with a substantial and significant survival benefit.

Also, there was no information available on whether a resection was performed as an emergency procedure. This might be a limitation of our analyses, since an earlier study on the topic from Germany using comprehensive hospital billing data [37] found lower rates of emergency procedures in hospitals with larger caseloads. Given the association of larger hospital size and certification status, this finding might be partly transferable to our study setting. However, the share of emergency procedures in the earlier study ranged between 30.7% for very low-volume and 28.1% for very high-volume hospitals, indicating a rather moderate difference. Moreover, it is known that reasons for surgery performed as emergency procedure are, e.g., more advanced tumor stages, lymphatic invasion, venous invasion, and higher age [38]. By adjusting for all of these factors, we are confident that we at least partially adjusted for the adverse effect of emergency procedures.

Therapy modalities like, e.g., the TME technique or the application of postoperative chemo protocols were deliberately excluded in the multivariable regression models, since the realization of guideline-adherent therapy might be a characteristic of certified centers and, thus, a reason for better survival after treatment at these institutions. However, delving deeper into the differences between therapy pathways of center- and non-center patients would definitely be an interesting topic for future research. To achieve reliable and valid results, the WiZen study followed a conservative approach, which might have led to an underestimation of the center effect. Patients treated in a hospital that forms part of an association with a GCS-certified cancer center were regarded as center patients, although their treatment might still not have met center standards. Furthermore, patients treated in hospitals that later became certified were allocated to the group treated in non-certified hospitals.

## 5. Conclusions

The results of the WiZen study contribute to a continuously growing international evidence base pointing towards the additional value of cancer treatment in designated centers that are defined not only by their caseload, but also by other quality indicators like guideline adherence. It represents the largest study to date on the efficacy of certification programs in the German health care system. The presented results show that treatment at certified colorectal cancer centers is associated with significant and substantial long-term benefits concerning overall and recurrence-free survival. In early-stage colon cancer, treatment at a certified institution contributes to lowering the overall mortality risk by more than 25%. This important information should be widely distributed to patients, referring outpatient physicians, and decision makers.

## Figures and Tables

**Figure 1 cancers-15-04568-f001:**
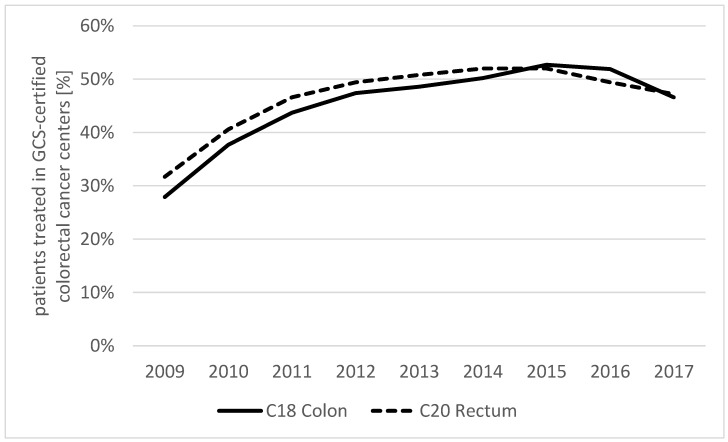
Share of patients treated in GCS-certified colorectal cancer centers according to diagnosis.

**Figure 2 cancers-15-04568-f002:**
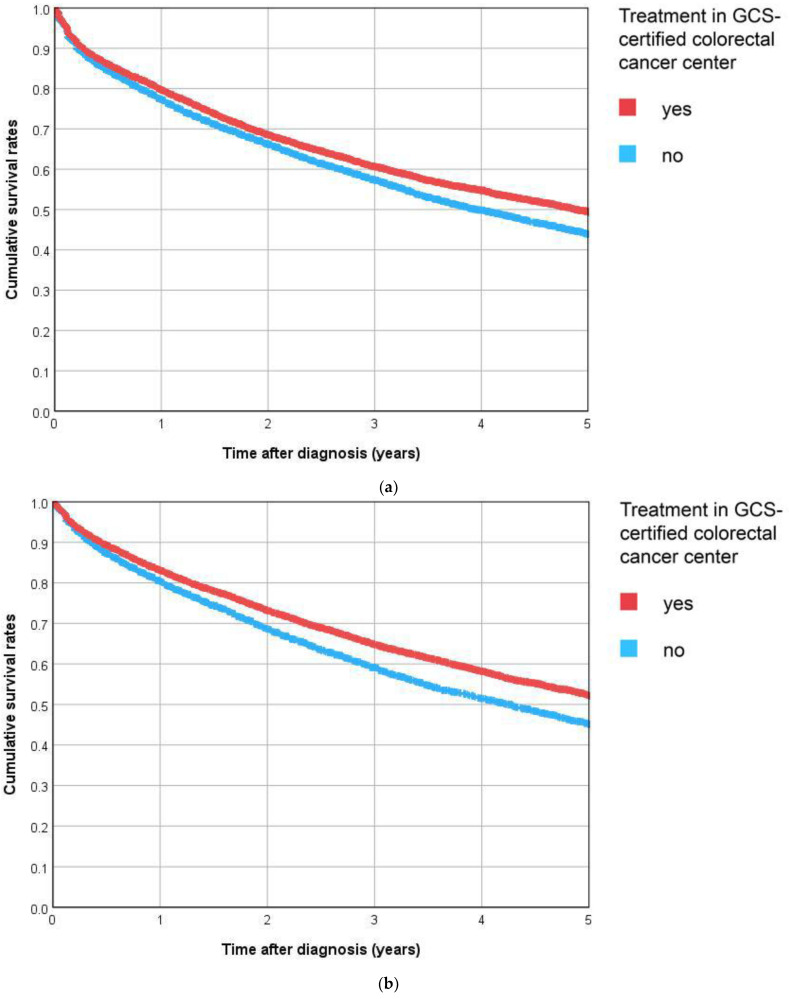
(**a**) C18 Colon, (**b**) C20 Rectum. Kaplan–Meier curves for overall survival according to diagnosis and treatment status.

**Figure 3 cancers-15-04568-f003:**
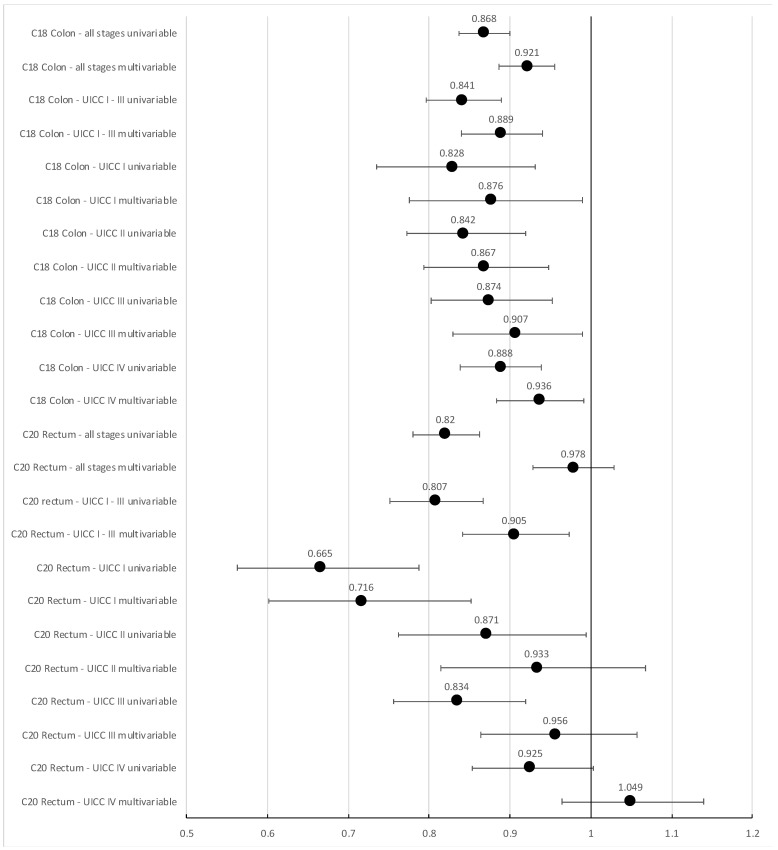
Unadjusted and adjusted (adjusted for age, sex, year of diagnosis, UICC stage—if not stratified for, grade, and lymphatic and venous invasion) hazard ratios with 95% CI for all-cause mortality following treatment in GCS-certified colorectal cancer centers compared to treatment in non-certified hospitals, stratified for location and UICC stage.

**Figure 4 cancers-15-04568-f004:**
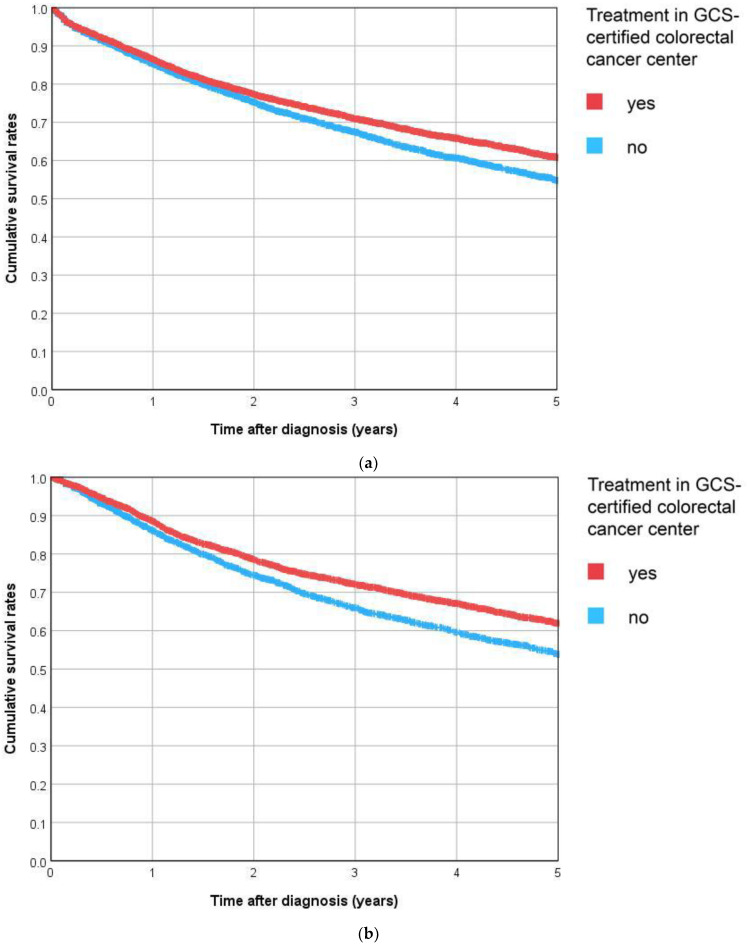
(**a**) C18 Colon, (**b**) C20 Rectum. Kaplan–Meier curves for recurrence-free survival in R0-resected patients with UICC stage I–III according to diagnosis and treatment status.

**Table 1 cancers-15-04568-t001:** Patient characteristics according to diagnosis and treatment status.

	C18 Colon				C20 Rectum			
Treatment in GCS-Certified Centers	Yes		No		Yes		No	
	n	%	n	%	n	%	n	%
Sex								
Female	6043	43.9	7515	44.9	2650	33.5	3353	37.1
Male	7717	56.1	9222	55.1	5254	66.5	5686	62.9
Age								
18–49	915	6.6	765	4.6	574	7.3	481	5.3
50–59	1801	13.1	1918	11.5	1553	19.6	1611	17.8
60–69	3086	22.4	3569	21.3	2136	27.0	2333	25.8
70–79	4860	35.3	6353	38.0	2480	31.4	3047	33.7
80+	3098	22.5	4132	24.7	1161	14.7	1567	17.3
Year of diagnosis								
2009–2011	3725	27.1	6504	38.9	2246	28.4	3409	37.7
2012–2014	4883	35.5	5136	30.7	2878	36.4	2793	30.9
2015–2017	5152	37.4	5097	30.5	2780	35.2	2837	31.4
Total	13,760	100.0	16,737	100.0	7904	100.0	9039	100.0

**Table 2 cancers-15-04568-t002:** Tumor characteristics according to diagnosis and treatment status.

	C18 Colon				C20 Rectum			
Treatment in GCS-Certified Centers	Yes		No		Yes		No	
	n	%	n	%	n	%	n	%
UICC stage								
I	2622	19.1	2915	17.4	1348	17.1	1511	16.7
II	3694	26.8	4190	25.0	1348	17.1	1577	17.4
III	2892	21.0	3352	20.0	2913	36.9	2770	30.6
IV	3634	26.4	3344	20.0	1747	22.1	1718	19.0
X	918	6.7	2936	17.5	548	6.9	1463	16.2
Grade								
G1	742	5.4	1232	7.4	400	5.1	683	7.6
G2	8913	64.8	10,286	61.5	5431	68.7	5704	63.1
G3/4	3059	22.2	3932	23.5	1127	14.3	1650	18.3
GX	1046	7.6	1287	7.7	946	12.0	1002	11.1
Lymphatic invasion								
L0	6783	49.3	7726	46.2	4297	54.4	3999	44.2
L1	4711	34.2	5905	35.3	1705	21.6	2304	25.5
LX	2266	16.5	3106	18.6	1902	24.1	2736	30.3
Vein invasion								
V0	9405	68.4	11,075	66.2	5157	65.2	5251	58.1
V1/2	1905	13.8	2401	14.3	779	9.9	1000	11.1
VX	2450	17.8	3261	19.5	1968	24.9	2788	30.8
Total	13,760	100.0	16,737	100.0	7904	100.0	9039	100.0

**Table 3 cancers-15-04568-t003:** Adjusted hazard ratios with 95% CI for recurrence-free survival (CCR data, UICC stage I–III, R0 only) following treatment in GCS-certified colorectal cancer centers, ref.: treatment in non-certified hospitals.

	HR	Lower 95% CI	Upper 95% CI
C18 Colon all stages univariable	0.845	0.802	0.891
C18 Colon all stages multivariable *	0.878	0.832	0.927
C20 Rectum all stages univariable	0.786	0.732	0.844
C20 Rectum all stages multivariable *	0.856	0.796	0.921

* adjusted for age, sex, year of diagnosis, UICC stage, grade, and lymphatic and venous invasion.

## Data Availability

The authors confirm that the data utilized in this study cannot be made available in the manuscript, the supplemental files, or in a public repository due to German data protection laws (‘Bundesdatenschutzgesetz’, BDSG).

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
