# Peer review of "Treatment of Colorectal Cancer in Certified Centers: Results of a Large German Registry Study Focusing on Long-Term Survival"

_cancers, 2023, doi:10.3390/cancers15184568_

Round 1

Reviewer 1 Report

based on this study,47.440 colorectal cancer patients treated between 2009 and 2017, Colorectal cancer patients treated in GCS-certified centers showed a better survival. 

Overall, this study will provide  evidence for clinical treatment of colorectal cancer  . There are also some issues require careful considerations.

1. Kaplan-meier and COX model is not suitable in survival analysis of colorectal cancer with long term survival and non-cancer death. competition risk model might be more suitable.

2. what is the biggest difference of treatment between GCS centers and other cencters

3.Medical costs and insurance should be listed

4.Survival of colon cancer diagnosis at 2015-2017 was significantly worse than survival of colon cancer diagnosis at 2009-2011, what is the reason.

Minor editing of English language required

Reviewer 2 Report

1. lines 306-307: I believe the 28% improvement in OS that you report is not correct. Where did this value come from?

2. The median and IQ values for follow up should be reported. It is important to statistically compare median follow up of patients treated at the certified vs non certified centers for each of the comparisons you make (eg stage I-III, all stages etc).

3. The discussion summarizes nicely the current literature which is consistent with your findings. However, this consistency makes your study less novel. What is the value that your study adds to current knowledge on this topic? This has to be apparent in discussion.

4. Can you delve into the nuances of these differences between designated centers and non certified centers? For example, how widely was TME implemented in the different centers? Do non certified centers use different protocols for chemo? Can you also do separate analyses for stage I colon cancer as it does not require chemo and might be a good example to see if differences in OS persist in this case.

5. As you acknowledged, it is very important to control for confounders. The teaching status of a hospital and the volume of surgical cases are two potential confounders as they might be more common in the certified centers group and in turn might be the reasons why outcomes are better in those hospitals and not the quality standards posed by certification. 

6. In addition, patient selection might differ across centers. I would expect you to adjust in multivariable analysis for more prognostic factors such as: T stage, number of nodes and receipt of postop chemo. All are known prognostic factors and have been included in the most recent MSK nomogram for colon cancer.

7. Can you also report KM plots and mutlivariable analysis results for RFS instead of OS alone?

Reviewer 3 Report

Comprehensive body of work, specific to Colorectal cancer in agreement with International literature.

However, No mention on emergency surgery which is up to 25% of presentations and tend to present to the local hospital, usually uncertified, and is known to produce less survival rates.

Reviewer 4 Report

This is a retrospective cohort study assessing whether treatment at German Cancer Society (GCS) certified centres is associated with improved overall survival of patients with colorectal cancer.

The study is interesting because, although suspected, there is little data available in the literature to confirm the above statement.

The introduction should elaborate more on what accreditation of a centre for the treatment of patients with colorectal cancer consists of.

Regarding the inclusion criteria (lines 111-128) please indicate whether patients with other non-colonectal cancer diagnoses were included.

It would be advisable to indicate how many centres not accredited in 2009 were accredited during the observation period of the study, since as suggested in the study limitations, this could affect the results.

The comments in the conclusions regarding adherence to clinical guidelines (line 301), lower cost of treatment in accredited centres (lines 304) and distance to centres (305-306) should be moved to the discussion section.

Conclusions should be adjusted to the analyses and results of the present study.

Round 2

Reviewer 1 Report

The author appropriately answered and modified the problems pointed out.

Reviewer 2 Report

The authors revised appropriately.

Minor editing is needed.